# The 16S rRNA Gene Sequencing of Gut Microbiota in Chickens Infected with Different Virulent Newcastle Disease Virus Strains

**DOI:** 10.3390/ani12192558

**Published:** 2022-09-24

**Authors:** Lina Tong, Wen Wang, Shanhui Ren, Jianling Wang, Jie Wang, Yang Qu, Fathalrhman Eisa Addoma Adam, Zengkui Li, Xiaolong Gao

**Affiliations:** 1College of Agriculture and Animal Husbandry, Qinghai University, Xining 810016, China; 2College of Veterinary Medicine, Northwest A & F University, Yangling, Xianyang 712100, China; 3State Key Laboratory of Plateau Ecology and Agriculture, Qinghai University, Xining 810016, China; 4State Key Laboratory of Veterinary Etiological Biology, Key Laboratory of Animal Virology of Ministry of Agriculture, Lanzhou Veterinary Research Institute, Chinese Academy of Agricultural Science, Lanzhou 730046, China; 5Mingzhou Regional Agriculture and Animal Husbandry Comprehensive Service Station of Suide Country, Suide, Yulin 718000, China

**Keywords:** Newcastle disease virus, gut microbiota, chicken, 16S rRNA gene

## Abstract

**Simple Summary:**

Newcastle disease (ND), which is caused by virulent Newcastle disease virus (NDV), is one of the most important viral diseases for chickens and birds. However, the intestinal pathogenesis of NDV is still poorly understood. To preliminarily investigate its intestinal pathogenesis mechanisms from the aspect of gut microbiota, the 16S rRNA gene sequencing technology was used to evaluate the gut microbiota composition changes post different virulent NDV infection. Results showed that different virulent NDV infection resulted in a different alteration of the gut microbiota in chickens, including a loss of probiotic bacteria and an expansion of some pathogenic bacteria. The above results suggest that NDV strains with different virulence have different impacts on chicken gut microbiota.

**Abstract:**

Newcastle disease virus (NDV) which is pathogenic to chickens is characterized by dyspnea, diarrhea, nervous disorder and hemorrhages. However, the influence of different virulent NDV strain infection on the host gut microbiota composition is still poorly understood. In this study, twenty 21-day-old specific pathogen free (SFP) chickens were inoculated with either the velogenic Herts33 NDV strain, lentogenic La Sota NDV strain or sterile phosphate buffer solution (PBS). Subsequently, the fecal samples of each group were collected for 16S rRNA sequencing. The results showed that the gut microbiota were mainly dominated by *Firmicutes*, *Bacteroidetes* and *Proteobacteria* in both healthy and NDV infected chickens. NDV infection altered the structure and composition of gut microbiota. As compared to the PBS group, phylum *Firmicutes* were remarkably reduced, whereas *Proteobacteria* was significantly increased in the velogenic NDV infected group; the gut community structure had no significant differences between the lentogenic NDV infected group and the PBS group at phylum level. At genus level, *Escherichia-Shigella* was significantly increased in both the velogenic and lentogenic NDV infected groups, but the *lactobacillus* was only remarkably decreased in the velogenic NDV infected group. Collectively, different virulent strain NDV infection resulted in a different alteration of the gut microbiota in chickens, including a loss of probiotic bacteria and an expansion of some pathogenic bacteria. These results indicated that NDV strains with different virulence have different impacts on chicken gut microbiota and may provide new insights into the intestinal pathogenesis of NDV.

## 1. Introduction

All vertebrate animals are inhabited by an immense population of microorganisms. The intestinal tracts maintain a particular rich and diverse microbial community numbering over trillions and with more than 1000 species [1,2]. This amazing amount of gut microbes were previously thought to be mainly beneficial for food sources utilization. Recently, with the development of research, scholars have found that these microbes also play an essential role in many aspects of the host’s physiology, including nutrient digestion, immune system development, detoxification of some compounds, and resistance to pathogens [3,4,5,6]. Although the diversity, roles and importance of these microbes in animal physiology have been illustrated, the biological significance of the presence of intestinal microbes in animals remains largely unclear. As the unique life history traits of birds are different from other vertebrates, such as hatching from eggs, chickens are an interesting study object for intestinal microbes. However, research of the avian intestinal microbiota was thought to have fallen behind that of other vertebrates, and recent study about avian intestinal microbiota mainly focused on composition of gut microbiota at different developmental stages, different segments of gut, and different living conditions. Little is known about the interaction between viral infection and avian gut microbiota. Existing reports were limited to avian influenza virus, infectious bronchitis, marek’s disease virus, infectious bursal disease virus and Newcastle disease virus [7,8,9,10,11,12,13,14]. With the ongoing prohibition of using antibiotics as a growth promoter and with the recognition of the beneficial role that a healthy gut microbiota plays in the promotion of growth and the resistance of viral and bacterial diseases [15,16], extensive study is still required to understand more about the interaction between virus infection and gut microbiota.

As a highly contagious avian disease, Newcastle disease (ND) causes hemorrhages and necrosis of the respiratory tract and the digestive tract resulting in high morbidity and mortality in chickens, and this has caused great economic losses to the poultry industry. Newcastle disease virus (NDV), the causative agent of ND, belongs to the family *Paramyxoviridae* and has a single-stranded, non-segmented, negative-sense RNA genome. Its genome is approximately 15.2 kb in length and contains six genes in the order of 3′-NP-P-M-F-HN-L-5′ [17]. According to the disease severity in chicken post infection, NDV strains are categorized as three pathotypes: highly pathogenic (velogenic) strains, which exhibit systemic infections with high mortality, including intestinal symptoms; intermediate (mesogenic) strains, which show intermediate virulence; and low-pathogenic (lentogenic) strains, which cause mild or asymptomatic infections that are restricted to the respiratory tract [18]. La Sota is a naturally occurring lentogenic NDV strain. Because of its good safety and efficacy, it has been widely used as a live vaccine to prevent Newcastle disease outbreaks in poultry practice. Herts33 is a velogenic strain isolated from fowl in 1933. Previously, Cui et al. reported that lentogenic NDV infection interferes with the formation of intestinal microbiota in newly hatched chicks by 16S rRNA gene sequencing technology [13]. However, whether or not the impact of different virulent NDVs on chicken gut microbiota are different is still unknown and needs further investigation. Here, we evaluated the influence of different virulent NDV strains on gut microbiota composition in 21-day-old specific pathogen free chickens by 16S rRNA sequencing technology. To our knowledge, this is the first report that illustrates the impact of different virulent NDV strains on chicken gut microbiota. 

## 2. Materials and Methods

### 2.1. Viruses

NDV strains La Sota and Herts33 were used in the present study. The La Sota strain is a class II genotype II lentogenic strain, and the Herts33 is a class II genotype IV virulent strain. These two strains were propagated in the allantoic cavity of 9–11 day-old embryonated specific pathogen-free (SPF) chicken eggs. Allantoic fluid was harvested from chicken embryos and stored at −70 °C. The virus median tissue culture infective dose (TCID_50_) was tested on DF-1 cell by Reed-Muench method [19].

### 2.2. Ethical Statement

The experiments were performed in strict accordance with Animal Ethics Procedures and Guidelines of the Ministry of Health in China and the ARRIVE guidelines. All experimental procedures were approved and supervised by the Ethics Committee for the Care and Use of Laboratory Animals in Qinghai University, China. Informed consent was obtained from the Jinan Sais Poultry Co., Ltd. (Jinan, China) in advance.

### 2.3. Experiment Design

Twenty 2-week-old specific pathogen free white Leghorns chickens were purchased from Jinan Sais Poultry Co., Ltd. The chickens were maintained in bio-security isolation units with feed and water administered *ad libitum*. After acclimatizing for one week, all chickens were divided into three groups with seven birds in two experiment groups and six birds in control group, namely group Herts33 (n = 7), group La Sota (n = 7) and group PBS (n = 6). Each bird in groups Herts33 and La Sota was challenged with 10^5^ TCID_50_/100 μL of the Herts33 strain or La Sota strain via eye drop (50 μL) and intranasal (50 μL) routes (EI/IN), respectively. Birds in the PBS group were challenged with 100 μL of PBS. All birds were monitored daily for clinical signs (depression, respiratory signs, diarrhea, etc.) and mortality. Cloacal swabs were used to collect about 200 mg of fecal sample from each bird in La Sota and PBS group at 4 days post challenge, and in Herts 33 group at 3 to 5 days post challenge when birds died for fecal DNA isolation. 

### 2.4. DNA Extraction and Library Construction

Total genomic DNA was extracted from about 200 mg collected feces using QIAamp 96 PowerFecal QIAcube HT kit (QIAGEN) following the manufacturer’s instructions. The concentration and purity of extracted DNA was verified with NanoDrop and agarose gel. Then the genome DNA was used as a template to amplify V3-V4 variable regions of 16S rRNA genes with universal primers 343F 5′-(TACGGRAGGCAGCAG)-3′ and 798R 5′-(AGGGTATCTAATCCT)-3′ and Tks Gflex DNA Polymerase (Takara). PCR were carried out in a 30 μL reaction mixture containing 2× Gflex PCR buffer 15 μL, primer 343F (5 pmol/μL) 1 μL, primer 798R (5 pmol/μL) 1μL, Tks Gflex DNA Polymerase 0.6 μL, and 50 ng DNA template. The PCR conditions were initial denaturation at 94 °C for 5 min, followed by 26 cycles of denaturation at 94 °C for 30 s, annealing at 56 °C for 30 s and extension at 72 °C for 20 s, with a final extension phase at 72 °C for 5 min. The PCR products were visualized using gel electrophoresis and purified with AMPure XP beads (Agencourt). The purified first round PCR product was used as a template to conduct second round PCR with the index primer pairs adapter I5 primer and adapter I7 primer. The PCR reaction system was carried out in a 30 μL reaction mixture as the first round PCR. The PCR condition were the same as the first round PCR except for the cycles were reduced to seven. After purification with the AMPure XP beads again, the final amplicon was quantified using Qubit dsDNA assay kit. Equal amounts of purified amplicon were pooled for subsequent sequencing using Illumina MiSeq system by oebiotech (Shanghai, China).

### 2.5. Bioinformatic Analysis

Raw sequencing data were in FASTQ format. Paired-end reads were then preprocessed using Trimmomatic software to detect and cut off ambiguous bases (N). We also cut off low quality sequences with an average quality score below 20 using a sliding window trimming approach. After trimming, paired-end reads were assembled using FLASH software. Assemble parameters were: 10 bp of minimal overlapping, 200 bp of maximum overlapping and 20% of maximum mismatch rate. To obtain high-quality clean reads, reads with ambiguous, homologous sequences or below 200 bp were abandoned. Reads with 75% of bases above Q20 were retained. Then, reads with chimera were detected and removed. These two steps were achieved using QIIME software (version 1.8.0). Then the clean reads were clustered to generate operational taxonomic units (OTUs) using VSEARCH software with 97% similarity. The representative read of each OTU was selected using the QIIME package. All representative reads were annotated and blasted against Silva database Version 123 using a RDP classifier (confidence threshold was 70%). Based on the OTUs information, an R package VennDiagram was performed to complete the Venn diagram. QIIME software was used for alpha and beta diversity analysis. The microbial diversity in samples was estimated using the alpha diversity that includes Chao1 index [20] and Shannon index [21]. The Bray-Curtis distance matrix performed by R package was used for Bray-Curtis Principal coordinates analysis (PCoA) to estimate the beta diversity. ANOVA and student’s *t*-test were performed to examine significant differences between various groups. Differences between groups were declared significant at *p* < 0.05. In the present study, all sequences have been deposited to the National Center for Biotechnology Information (NCBI) database under accession number PRJNA700718.

## 3. Results

### 3.1. Clinical Symptoms Post NDV Challenge

All chickens in the Herts 33 group showed overt clinical signs and died within 3–5 days. The clinical signs include severe depression, deep green and faint yellow diarrhea and nervous signs such as wing drop and leg paralysis, respiratory distress with gasping and sneezing, and dying 3–5 days post challenge. No chickens in the La Sota group or the PBS group exhibited clinical signs post challenge, except one in the La Sota group that showed a transient mild depression.

### 3.2. Sequencing Results Overview

In the present study, twenty fecal samples (seven Herts33 challenged, seven La Sota challenged, six PBS negative control) were collected and processed for 16S rRNA gene sequencing and analysis. After quality control, about 60,617 to 72,639 clean reads were obtained. The valid reads were distributed between 46,932 and 69,273 post removing chimera. The average length of valid reads is 406.47 to 425.62 bp and the OTU number of each sample was distributed between 471 and 1477. The Good’s coverage ranged from 99.32% to 99.59%, indicating a good sequencing depth enough to cover the majority of the gut microbiota in each sample.

### 3.3. A Decrease in the Microbial Diversity in Gut Microbiota with NDV Infection

A Venn diagram reveals the shared and specific OTUs among the different groups. As shown in Figure 1, the average number of observed OTUs in the NDV infected chicken samples was more than that of the PBS control group (Figure 1 Venn plot. Herts 2227, La Sota 1861, PBS 1802). 

To evaluate the influence of NDV infection on gut microbiota diversity and richness, the Chao, Shannon and Simpson indices were calculated, and these indices of each sample are shown in Table 1. As shown in Figure 2, the average Chao index of the Herts33, La Sota and PBS group was 834.884, 878.283 and 725.671, respectively. The average Shannon index of the Herts33, La Sota and PBS group was 1.358, 2.431 and 2.4, respectively. The average Simpson index of the Herts33, La Sota and PBS group was 0.27, 0.59 and 0.69, respectively. A relative higher Chao index, Shannon index and Simpson index means a higher richness and diversity of the bacteria. However, a Wilcox test showed that the Chao index, Shannon index and Simpson index had no significant differences from each other (Wilcox, *p* > 0.05), suggesting that the bacterial richness and diversity were not affected by NDV. 

### 3.4. Gut Bacterial Beta-Diversity Analysis

To analyze the similarities and differences of bacterial communities among these three groups of chickens, the Bray–Curtis similarity was calculated. The Bray–Curtis based analysis of similarities indicated that the microbiota among the three groups were significantly different from each other (R = 0.3908, *p* = 0.001). Furthermore, principal coordinate analysis (PCoA) was performed based on Bray–Curtis distances to visualize the similarity of the microbial community structure in different groups. As shown in Figure 3, PC1 and PC2 account for 33.23 and 18.89% of the total variation. There was distinguishing clustering of the samples from each group. However, partial samples from La Sota and PBS were close to each other. The PCoA result suggested distinct differences in the bacterial composition among the three groups.

### 3.5. NDV Infection Alter the Gut Microbiome Composition in Chickens

To elucidate the effect of NDV infection on gut bacterial composition, we evaluated the gut microbiota at different taxonomical levels. The overall bacterial composition of different groups at the phylum level was shown in Figure 4A,a; sequences that accounted for very small proportions were combined as others. From Figure 4A,a, we found that *Firmicutes*, *Proteobacteria* and *Bacteroidetes* were the three most abundant phyla in all groups. The average relative abundance of phylum *Firmicutes* in the Herts33 group was significantly lower than that in the other two groups (Figure 5a, ANOVA *p* < 0.01), while the relative abundance of phylum *Proteobacteria* was significantly higher than that in the La Sota and PBS groups (Figure 5a, ANOVA, *p* < 0.01).

When analyzed at the genus level, as shown in Figure 4B,b, the main genera in these three groups included *lactobacillus*, *Escherichia-Shigella*, *Enterococcus* and *Bacteroides*. The top 10 significantly different genus were *lactobacillus*, *Escherichia-Shigella*, *enterococcus*, *GCA-900066575* (*an uncultured Clostridium* sp.), *Clostridium*, *Pseudomonas*, *Azospirillum*, *Pseudogracilibacillus*, *Weissella* and *Brachybacterium* (Figure 5b, ANOVA, *p* < 0.05). The relative abundance of genus *lactobacillus* in the Herts33 group was significantly lower than the other two groups. The relative abundance of genus *Escherichia-Shigella* in the Herts33 group and the La Sota group was significantly higher that of the PBS group (Figure 6a,b, *t*-test, *p* < 0.05). However, the abundance of genus *lactobacillus* had no significant difference between the La Sota group and the PBS group (Figure 6b, *t*-test *p* > 0.05). The relative abundance of genus *enterococcus* in the Herts33 group and the La Sota group was significantly lower that of the PBS group.

## 4. Discussion

The intestine tract of chickens, as of other animals, is populated with a relatively rich and diverse microbial community, including bacteria, viruses, fungi and protozoa. These incredibly complex microbial community possess important functions for their hosts in many aspects. At the same time, the intestinal microbiota is dynamic and influenced by environment, diet, age, antibiotics, pathogen infection and other factors [22]. The maintenance of a health gut microbiota, therefore, is very important and contributes significantly to the overall health and performance of a flock [23]. If the structure and composition of gut microbiota is disturbed, this may have a severe impact on the chickens’ growth performance and may enhance the risk for systemic diseases including infectious diseases [24]. Viruses and bacteria could interact with each other in the gut and, thus, affect the virus replication and transmission [25,26]. Therefore, this study was designed to evaluate whether NDV infection could cause the alteration of chicken gut microbiota.

A previous study reported that the infection of NDV resulted in the disproportion of intestinal microbiota [13]. In the present study, we compared the gut microbiota between different virulent NDV strain infected chickens and non-infected chickens by 16S rRNA gene sequencing and found that NDV could alter the gut microbiota composition at different levels, which is in line with previous observations [13]. To examine whether vertical infection of NDV influence the formation of the intestinal community, Cui evaluated the effect of NDV infection on chick embryos at hatch. Their result showed that NDV infection decreased the richness and overall diversity of duodenal flora, but the richness and diversity of cecal microflora was not affected. Our result is in accordance with Cui’s result on cecal as the alpha diversity indexes were not significantly different between NDV infection groups and the control group in the present work [13]. The results of PCoA indicated that the NDV infection altered the structure of gut microbiota, which is consistent with the results of the previous study [13]. From the PCoA results we concluded that different virulent NDV infections have varying influence on chicken microbiota. 

Many studies have demonstrated that chicken gut microbiota consists of three major bacterial phyla, namely the *Firmicutes*, the *Proteobacteria*, and the *Bacteroidetes*. Our present study also found that the above three bacterial phyla were the predominant observed bacterial taxa, which confirmed previous observations [27]. The functions of *Firmicutes* and *Bacteroidetes* are closely related with carbohydrate and protein metabolism and play a role in energy production [28,29]. At the same time, some members in phyla *Firmicutes* could regulate the inflammation by induction of anti-inflammatory cytokines [30]. As a minor constituent in the fecal microbial community, the *Proteobacteria* accounted for only 2.31% in the PBS group (Figure 5), and this group included many pathogenic bacteria, such as *Escherichia*, *Shigella*, *Salmonella*, *Clostridium cluster* XI, *Vampirovibrio* and so on [31]. As compared to the PBS group, the increase of the *Proteobacteria* and decrease of *Firmicutes* in two NDV infected groups (velogenic Herts33 VS PBS, *p* < 0.01; lentogenic La Sota VS PBS, *p* > 0.05) may be a sign of disease in chickens. 

*Lactobacillus* are one of the predominant bacterial genera in the gastrointestinal tract of chicken [27], and provide great benefits for chickens, such as help in carbohydrate fermentation and restriction of the replicate of other bacteria species by production of lactate, bacteriostatic and bactericidal substances [32,33,34,35]. In addition, *lactobacillus* could modulate the immune system, and significant enhancement of the immune response was also observed in chickens [36]. Now, *lactobacillus* strains are actually considered as safe organisms and have been widely used as a probiotics to improve growth performance and inhibit the potential pathogenic microorganisms such as Salmonella and *Escherichia-coli* [37,38]. In this study, as the most abundant genus and top different genus, the relative abundance of *Lactobacillus* in velogenic NDV infection group was significantly lower than that of PBS group (*t*-test, *p* < 0.01), but that has no significant differences between the lentogenic NDV infection group and the PBS group (*t*-test, *p* > 0.05). The decline of *Lactobacillus* was also observed in chickens post Eimeria tenella or H9N2 avian influenza virus infection [8,39]. It has been shown that some Lactobacillus can enhance the IFN and IL-22 production and response [40,41], and higher abundance of the *Lactobacillus* was associated with restoration of the epithelial barrier integrity [42,43]. Furthermore, oral administration of *Lactobacillus* can effectively relieve diarrhea by regulating intestinal microflora and improving immune system function [44]. As a result, we speculated that the differences in the abundance of *Lactobacillus* post different virulent NDV infections would account for different clinical signs in intestines. The detailed mechanisms of why different virulent NDV have different influence on the quantity of *Lactobacillus* and the exact pathway NDV uses to affect the *Lactobacillus* need further investigation.

In contrast, genera *Escherichia* and *Shigella*, to which also belong some pathogenic species or strains, were both significantly increased in the two NDV infected groups. In the velogenic NDV infected group, the average relative abundance of *Escherichia-Shigella* increased from 1.4% to 53.3%, while that increased from 1.4% to 19.8% in the lentogenic NDV infected group. The increase of *Escherichia-Shigella* was also observed in the infection of H9N2 avian influenza virus, ALV-J, duck reovirus and Eimeria tenella [45,46]. Additionally, some reports suggest a positive correlation between the abundance of *Escherichia-Shigella* and the development of necrotic enteritis in chickens [47]. Moreover, previous studies have found that IFN-α, IFN-β, IFN-γ, and IL-22 expression were negatively correlated with *Clostridium cluster* XI, *Escherichia*, and *Shigella* species post AIV infection [9,40]. In H9N2 AIV infected chickens, elevated levels of IFNs caused the dysbiosis of commensal gut microbiota and decreased the number of lactic-acid-producing bacteria due to an increased relative abundance of pathogenic *Proteobacteria*, including *Shigella*, which produce inflammation in GIT [48]. These data indicate that NDV infection might increase the possibility of subsequent infection by other pathogens. Whether or not the different expression levels of cytokines, such as IFNs, IL22, IL17, which were induced by NDV infection, account for differences in gut microbiota alteration and clinical symptoms post different virulent NDV strain infection needs further study.

## 5. Conclusions

In conclusion, our study demonstrated that significant dysbiosis occurs in the gut microbiota of chickens post NDV infection. The alteration of gut microbiota was dominated by an increased relative abundance of the genera *Escherichia* and *Shigella* and apparent decrease in the level of the *lactobacillus*. These observations indicate a fundamental alteration in the chicken gut microbiota post NDV infection. Further investigation of the mechanisms underlying these interactions could help reveal useful targets and treatment approaches for restoring the gut microbiota to help combat NDV.

## Figures and Tables

**Figure 1 animals-12-02558-f001:**
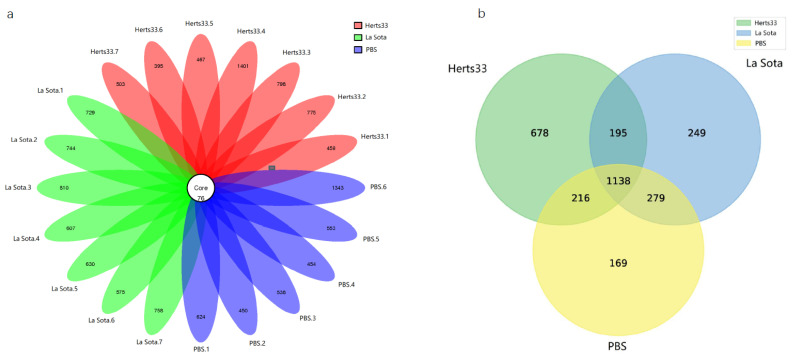
The community composition analysis. (**a**) Venn diagram showing overlap in OTUs of differential abundance of each sample in Herts33, La Sota and PBS groups. (**b**) Venn diagram showing overlap in OTUs of differential abundance in Herts33, La Sota and PBS groups.

**Figure 2 animals-12-02558-f002:**
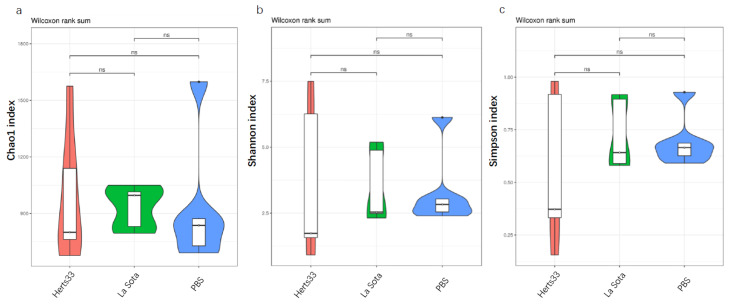
The microbial diversity index analysis. (**a**) Chao index. (**b**) Shannon index. (**c**) Simpson index. ns: no significant difference (*p* > 0.05).

**Figure 3 animals-12-02558-f003:**
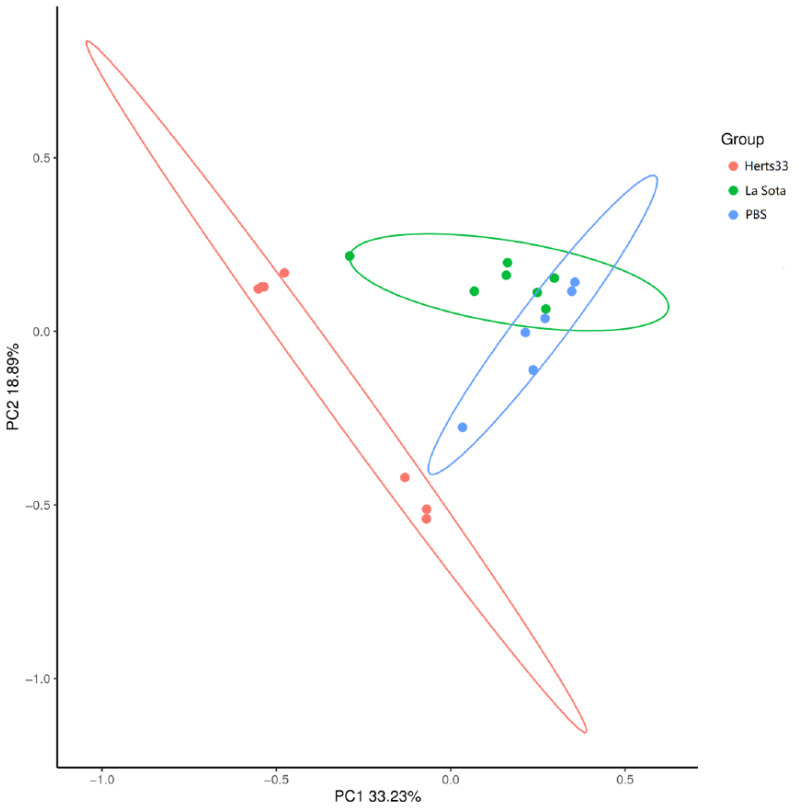
PCoA analysis of similarities between different groups. Principal component (PC) 1 and 2 accounted for 33.23% and 18.89% of the variance, respectively.

**Figure 4 animals-12-02558-f004:**
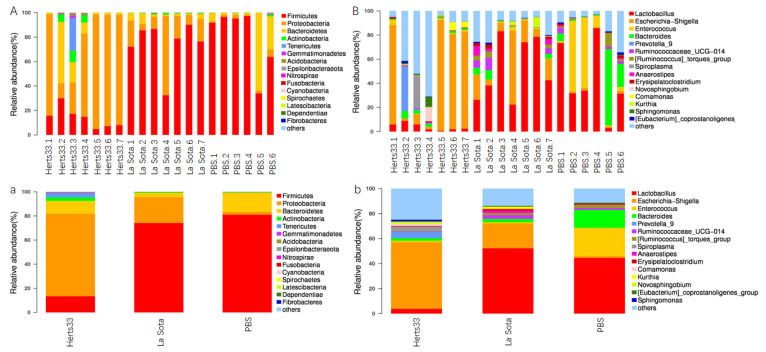
Microbial composition of different samples and groups. Each bar represents the relative abundance of each bacterial taxon within samples or groups. (**A**) Taxa assignments of each sample at Phylum level. (**a**) Taxa assignments of each group at Phylum level. (**B**) Taxa assignments of each sample at Genus level. (**b**) Taxa assignments of each group at Genus level.

**Figure 5 animals-12-02558-f005:**
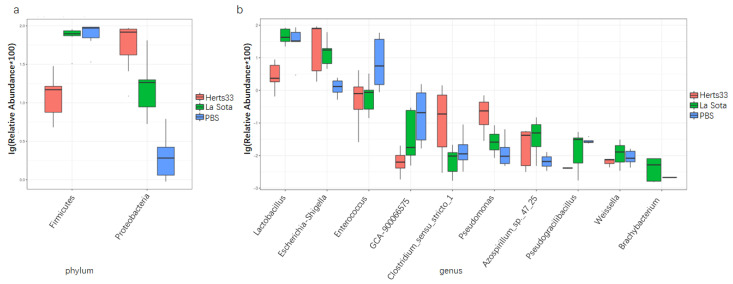
The relative abundance of the top 10 different bacteria in the three groups (ANOVA), expressed as an average percentage of the total: (**a**) At phylum level; (**b**) At genus level.

**Figure 6 animals-12-02558-f006:**
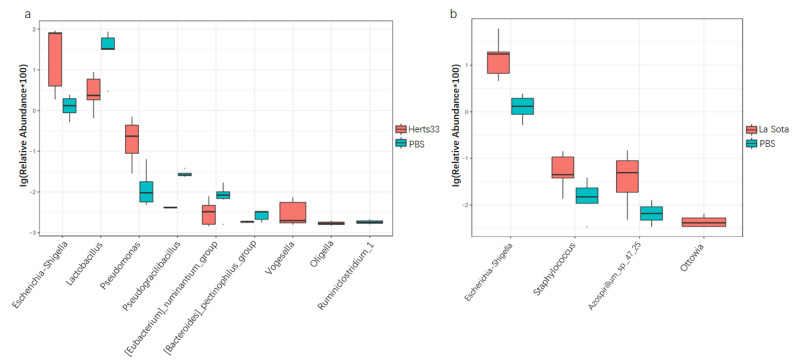
The relative abundance of the top 10 different genera bacteria between NDV challenge groups and the control group (*t*-test), expressed as an average percentage of the total: (**a**) Herts33 VS PBS; (**b**) La Sota VS PBS.

**Table 1 animals-12-02558-t001:** The estimators of sequence diversity and richness.

Samples	Reads	OTUs	Simpson	Chao1	Observed Species	Shannon	Goods Coverage	PD Whole Tree
Herts33.1	67,293	535	0.315108	764.6402	447.8	1.40745	0.995131	21.40172
Herts33.2	54,131	851	0.917592	1126.855	815.2	5.980166	0.995479	36.46119
Herts33.3	57,483	874	0.918702	1151.576	820.5	6.545797	0.995068	37.76043
Herts33.4	50,601	1477	0.980155	1575.387	1458.1	7.497044	0.995838	57.57711
Herts33.5	69,273	543	0.155358	759.3362	449.8	0.914995	0.995384	21.97474
Herts33.6	61,314	471	0.372028	677.053	416.3	1.730422	0.995881	20.37164
Herts33.7	63,697	579	0.349542	799.8134	501.5	1.730623	0.99504	24.97967
La Sota.1	58,778	805	0.913189	1006.878	746.1	5.120451	0.99454	30.27542
La Sota.2	58,034	820	0.9171	1049.569	756.2	5.189811	0.994462	30.44099
La Sota.3	61,316	886	0.584854	1022.178	815.2	2.510756	0.994423	35.28361
La Sota.4	64,931	683	0.594488	824.6888	609.2	2.538315	0.995507	27.8734
La Sota.5	62,199	706	0.580574	837.9335	634.3	2.316419	0.99486	27.89631
La Sota.6	60,679	651	0.641917	794.4215	593.1	2.462201	0.99524	27.51244
La Sota.7	57,813	834	0.877119	996.3602	774.2	4.652152	0.994438	31.2902
PBS.1	61,439	700	0.619337	861.2555	630.9	3.045674	0.995023	26.75791
PBS.2	59,118	526	0.647619	690.9104	476.7	2.510859	0.995636	21.37007
PBS.3	60,339	614	0.683144	876.2117	544.2	2.643193	0.994499	24.31326
PBS.4	62,201	530	0.687748	699.7192	463.5	2.40015	0.995292	22.87679
PBS.5	55,910	629	0.591706	812.8232	585.7	3.014565	0.995425	24.2154
PBS.6	46,932	1419	0.928459	1598.205	1411.9	6.129219	0.993234	51.03208

## Data Availability

All data generated or analyzed during this study are included in this published article and are also available from the corresponding author on reasonable request.

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
