# Peer review of "The 16S rRNA Gene Sequencing of Gut Microbiota in Chickens Infected with Different Virulent Newcastle Disease Virus Strains"

_animals, 2022, doi:10.3390/ani12192558_

Round 1
Reviewer 1 Report
Tong et al. present a manuscript describing their results of chicken gut microbiota analysis after infection with Newcastle disease virus of divergent pathogenicity. It is well known that NDV infection leads to gastro-intestinal clinical signs. This manuscript proves the impact of infection with a virulent and a ND vaccine strain onto bacterial composition.
Overall, the manuscript is written well and there are only minor indications to point out.
· Line 85: “low-pathogenic” instead of “apathogenic”
· Line 107: Please provide a reference for Reed-Muench method.
· Line 164: Please provide a reference regarding the program that was used for the statistical analyses and the generation of the graphs.
· Line 173: Were the Herts33 chickens already dead when sampled? In the MM section it is said that sampling was done day 4 after infection, but Herts33 chickens died between day 3 and 5 after infection. Please clarify!
· Line 184: Please explain what operational taxonomic units are or how they are defined.
· Line 187: Please correct “Hetts” to “Herts” throughout the whole manuscript.
· Lines 195-196: Please provide a reference for indices in MM and maybe shortly describe what they mean and express.
· Line 217: There should be a short explanation in MM section about PCoA. What kind of analysis is that, what is being done and what does PC1 and PC2mean. Otherwise figure 3 is not very comprehensible.
· Figure 4, 5 and 6: The letters must be taller. It is not well readable.
· Figure 4: What about the individual differences between the single animals? Herts 2, 3 and 4 are different from Herts 1, 5, 6, 7 and LaSota 1, 2 und 7 are different from the others LaSota animals. Was there a consensus with maybe clinical signs?
· Line 254: What does GCA-900066575 mean?
· Discussion: Just our of curiosity: Can you suggest how much time it would take until the gut microbiota regains its previous composition?
Author Response
The comments and suggestions of the reviewers were highly insightful and enabled us to greatly improve the quality of our manuscript. All issues mentioned in the reviewers' comments has been revised accordingly and revisions are marked in Red in the revised manuscript. In the following pages are our point-by-point responses to each of the comments and suggestions of the reviewers.
- Line 85: “low-pathogenic” instead of “apathogenic”
Thank you for your comments, we have revised this part according to your kind suggestions.
- Line 107: Please provide a reference for Reed-Muench method.
Thank you, we have added a reference for Reed-Muench method in the revised manuscript.
- Line 164: Please provide a reference regarding the program that was used for the statistical analyses and the generation of the graphs.
Thank you, we have added the program that was used for the statistical analyses and the generation of the graphs in materials and methods. And the part 2.5 and 2.6 were combined as one paragraph.
- Line 173: Were the Herts33 chickens already dead when sampled? In the MM section it is said that sampling was done day 4 after infection, but Herts33 chickens died between day 3 and 5 after infection. Please clarify!
Thank you for your professional comments. It is our mistake that did not clearly stated this sentence and we have revised this sentence in the revised manuscript.
- Line 184: Please explain what operational taxonomic units are or how they are defined.
Thank you for your comments. We have changed the state in this sentence in the revised manuscript. Operational Taxonomic Unit (OTU) is an artificially defined taxon. In phylogenetic or population genetic studies, a symbol was assigned to a taxon artificially for easy analysis. To understand the number of bacteria and genera in the sequencing results of a sample, it is necessary to cluster the sequences. Through the categorization operation, sequences are divided into many groups according to their similarity to each other. A group is an OTU. OTU classification is usually carried out with 97% sequence similarity.
- Line 187: Please correct “Hetts” to “Herts” throughout the whole manuscript.
Thank you, we have correct “Hetts” to “Herts” throughout the whole manuscript.
- Lines 195-196: Please provide a reference for indices in MM and maybe shortly describe what they mean and express.
Thank you, we have added reference for indices in materials and methods.
- Line 217: There should be a short explanation in MM section about PCoA. What kind of analysis is that, what is being done and what does PC1 and PC2 mean. Otherwise figure 3 is not very comprehensible.
Thank you, we have added a short explanation in materials and method section about PCoA.
- Figure 4, 5 and 6: The letters must be taller. It is not well readable.
Thank you. We have revised the letters in figure 4, 5 and 6.
- Figure 4: What about the individual differences between the single animals? Herts 2, 3 and 4 are different from Herts 1, 5, 6, 7 and LaSota 1, 2 und 7 are different from the others LaSota animals. Was there a consensus with maybe clinical signs?
Thank you for your comments. As figure 4 showed, the Herts 2, 3 and 4 are different from Herts 1, 5, 6, 7 and LaSota 1, 2 und 7 are different from the others LaSota animals. And the chickens numbered 1, 3, 4 and 5 showed symptoms at 3 days post challenge (dpc), chickens numbered 2 and 6 showed symptoms at 4dpc, chickens numbered 7 showed symptoms at 5dpc in Herts group. Chickens numbered 3 and 4 died at 3dpc, chickens numbered 1, 5, 6 died at 4dpc, chickens numbered 2 and 7 died at 5dpc. As in La Sota group, only chicken numbered 7 showed a transient mild depression during the experiment. So the direct correlation between clinical signs and alteration of the gut microbiome composition is not very good. This phenomenon may be due to the individual differences in animals and relatively small sample size in the present study. We will try to test this hypothesis with large samples in the future. Thank you for your professional suggestions.
- Line 254: What does GCA-900066575 mean?
Thank you. The GCA-900066575 is an uncultured Clostridium sp. And we have noted in the revised manuscript.
- Discussion: Just our of curiosity: Can you suggest how much time it would take until the gut microbiota regains its previous composition?
Thank you for your professional comments. It’s a difficult question to answer. As the degree of dysbiosis of gut microbiota post NDV infection is different, and the intestinal damage caused by NDV infection are also different, so it hard to evaluate how much time it would take until the gut microbiota regains its previous composition. But we speculate that the gut microbiota begin to regains its previous composition when the clinical symptom disappear post NDV infection, maybe one week to a month. If we have much more time and much more fund, it’s an interesting thing to investigate in the future. Thank you.
Reviewer 2 Report
Tong et al., here reported that different alteration of gut microbiota in chickens with velogenic and lentogenic Newcastle disease virus (NDV) using 16S rRNA gene sequencing, as evidenced by decreased probiotic bacteria and increased pathogenic bacteria in the gut microbiota of the velogenic NDV-infected chickens, but the alteration of the lentogenic NDV-infected gut microbiota is comparable to that in the mock-infected chickens. The design of this study is well-done, and the results are convinced. Some grammatical and wording errors need to be modified before accepted for publication.
Author Response
The comments and suggestions of the reviewers were highly insightful and enabled us to greatly improve the quality of our manuscript. All issues mentioned in the reviewers' comments has been revised accordingly and revisions are marked in Red in the revised manuscript. In the following pages are our point-by-point responses to each of the comments and suggestions of the reviewers.
Tong et al., here reported that different alteration of gut microbiota in chickens with velogenic and lentogenic Newcastle disease virus (NDV) using 16S rRNA gene sequencing, as evidenced by decreased probiotic bacteria and increased pathogenic bacteria in the gut microbiota of the velogenic NDV-infected chickens, but the alteration of the lentogenic NDV-infected gut microbiota is comparable to that in the mock-infected chickens. The design of this study is well-done, and the results are convinced. Some grammatical and wording errors need to be modified before accepted for publication.
Thank you very much for your kind suggestions. We have corrected the grammatical and wording errors in the revised manuscript.
Reviewer 3 Report
The manuscript of Tong et al investigated the gut microbiota changes in chickens infected with two NDV strains with different virulent properties along with non-infected chickens as a control group. The authors have used 6S rRNA sequencing to analyze changes in gut microbiota profiles. The finding of the study indicated that NDV with different virulence have different impact on chicken gut microbiota. The manuscript is well written and easy to understand. However, I have some concerns in the procedure as indicated below.
This study lacks quality control of the microbiota profiling protocol with negative and positive controls
Presence of contamination in DNA extraction kits and other laboratory reagents is a major concern in molecular-based studies of microbial communities. As those reagent contaminants are not easy to avoid, it is very important to have negative controls (DNA extraction blanks, Negative PCR controls) and proceed those negative controls though out the library preparation steps and include the respective libraries in the final library pool for sequencing.
In addition to that, most studies include commercially available positive control with a known mixture of bacteria (such as microbial community standard from Zymo-BIOMICS™, Zymo Research) during the DNA extraction phase and incorporate the respective positive control library in the final sequencing pool.
Contaminants are present in negative controls and as a foreign taxa in positive controls. Then based on two approaches we can identify and remove the contaminating sequences.
, , , et al. Laboratory contamination over time during low‐biomass sample analysis. Mol Ecol Resour. 2019; 19: 982– 996. https://doi.org/10.1111/1755-0998.13011
Glassing, A., Dowd, S.E., Galandiuk, S. et al. Inherent bacterial DNA contamination of extraction and sequencing reagents may affect interpretation of microbiota in low bacterial biomass samples. Gut Pathog 8, 24 (2016). https://doi.org/10.1186/s13099-016-0103-7
Davis, N.M., Proctor, D.M., Holmes, S.P. et al. Simple statistical identification and removal of contaminant sequences in marker-gene and metagenomics data. Microbiome 6, 226 (2018). https://doi.org/10.1186/s40168-018-0605-2
As the experiments are already done, I suggest therefore to look into the common laboratory contaminants based on the literature and asses if they have impacted the final outcome of the study.
2.5. Bioinformatic analysis
Need some more description of the bioinformatic analysis
Apart from chimeric sequence removing what are the other filtering steps performed to obtain high quality reads?
If so, please include those steps in the manuscript
What are the tools and software you used for downstream analysis of the resulting OTUS? i.e. alpha and beta diversity analysis
3.2. Sequencing results overview
Why you used the word “tags” for the reads resulting from amplicon sequencing?
Minor comments
Please use growth performances instead of grow performances throughout the manuscript.
Author Response
The comments and suggestions of the reviewers were highly insightful and enabled us to greatly improve the quality of our manuscript. All issues mentioned in the reviewers' comments has been revised accordingly and revisions are marked in Red in the revised manuscript. In the following pages are our point-by-point responses to each of the comments and suggestions of the reviewers.
The manuscript of Tong et al investigated the gut microbiota changes in chickens infected with two NDV strains with different virulent properties along with non-infected chickens as a control group. The authors have used 16S rRNA sequencing to analyze changes in gut microbiota profiles. The finding of the study indicated that NDV with different virulence have different impact on chicken gut microbiota. The manuscript is well written and easy to understand. However, I have some concerns in the procedure as indicated below.
This study lacks quality control of the microbiota profiling protocol with negative and positive controls
Presence of contamination in DNA extraction kits and other laboratory reagents is a major concern in molecular-based studies of microbial communities. As those reagent contaminants are not easy to avoid, it is very important to have negative controls (DNA extraction blanks, Negative PCR controls) and proceed those negative controls thoughout the library preparation steps and include the respective libraries in the final library pool for sequencing. In addition to that, most studies include commercially available positive control with a known mixture of bacteria (such as microbial community standard from Zymo-BIOMICS™, Zymo Research) during the DNA extraction phase and incorporate the respective positive control library in the final sequencing pool.
Contaminants are present in negative controls and as a foreign taxa in positive controls. Then based on two approaches we can identify and remove the contaminating sequences.
Weyrich, LS, Farrer, AG, Eisenhofer, R, et al. Laboratory contamination over time during low‐biomass sample analysis. Mol Ecol Resour. 2019; 19: 982–996. https://doi.org/10.1111/1755-0998.13011
Glassing, A., Dowd, S.E., Galandiuk, S. et al. Inherent bacterial DNA contamination of extraction and sequencing reagents may affect interpretation of microbiota in low bacterial biomass samples. Gut Pathog 8, 24 (2016). https://doi.org/10.1186/s13099-016-0103-7
Davis, N.M., Proctor, D.M., Holmes, S.P. et al. Simple statistical identification and removal of contaminant sequences in marker-gene and metagenomics data. Microbiome 6, 226 (2018). https://doi.org/10.1186/s40168-018-0605-2
As the experiments are already done, I suggest therefore to look into the common laboratory contaminants based on the literature and asses if they have impacted the final outcome of the study.
Thank you very much for your kind and professional suggestions. After consultation of the Oebiotech company (Shanghai, China. https://www.oebiotech.com/) responsible for this study, we were informed that their company have strict quality control systems that cover the entire process from DNA extraction, library construction to final sequencing. And each sample they tested has a negative control to eliminate the interference of environmental factors. And high-throughput 16S rRNA sequencing technology was a very mature technology widely used for gut microbiota study. So, we are confident in the results of the present article. We will pay special attention to this issue. Thank you very much again for your professional comments and suggestions.
2.5. Bioinformatic analysis
Need some more description of the bioinformatic analysis
Apart from chimeric sequence removing what are the other filtering steps performed to obtain high quality reads? If so, please include those steps in the manuscript.
Thank you, we have added bioinformatic analysis and other filtering steps in materials and methods in our revised manuscript.
What are the tools and software you used for downstream analysis of the resulting OTUS? i.e. alpha and beta diversity analysis
Thank you for you comments, we have added the software used for alpha and beta diversity analysis in materials and methods.
3.2. Sequencing results overview
Why you used the word “tags” for the reads resulting from amplicon sequencing?
Thank you for your professional suggestions. I am very sorry for an improper use of words and we have use “reads” instead of “tags” in the revised manuscript.
Minor comments
Please use growth performances instead of grow performances throughout the manuscript.
Thank you very much, we have use “growth performances” instead of “grow performances” in the revised manuscript.